# High-Caloric Diets in Adolescence Impair Specific GABAergic Subpopulations, Neurogenesis, and Alter Astrocyte Morphology

**DOI:** 10.3390/ijms25105524

**Published:** 2024-05-18

**Authors:** Bárbara Mota, Ana Rita Brás, Leonardo Araújo-Andrade, Ana Silva, Pedro A. Pereira, M. Dulce Madeira, Armando Cardoso

**Affiliations:** 1Unit of Anatomy, Department of Biomedicine, Faculty of Medicine, University of Porto, Alameda Prof. Hernâni Monteiro, 4200-319 Porto, Portugal; barbaramota@med.up.pt (B.M.);; 2NeuroGen Research Group, Center for Health Technology and Services Research (CINTESIS), Rua Dr. Plácido da Costa, 4200-450 Porto, Portugal; 3CINTESIS@RISE, Faculty of Medicine, University of Porto, Alameda Prof. Hernâni Monteiro, 4200-319 Porto, Portugal

**Keywords:** gamma-aminobutyric acid (GABA), high-caloric diet, high-fat, high-sugar, Western diet, hippocampus, neurogenesis, astrocytes, interneurons

## Abstract

We compared the effects of two different high-caloric diets administered to 4-week-old rats for 12 weeks: a diet rich in sugar (30% sucrose) and a cafeteria diet rich in sugar and high-fat foods. We focused on the hippocampus, particularly on the gamma-aminobutyric acid (GABA)ergic system, including the Ca^2+^-binding proteins parvalbumin (PV), calretinin (CR), calbindin (CB), and the neuropeptides somatostatin (SST) and neuropeptide Y (NPY). We also analyzed the density of cholinergic varicosities, brain-derived neurotrophic factor (*BDNF*), reelin (*RELN*), and cyclin-dependent kinase-5 (*CDK-5*) mRNA levels, and glial fibrillary acidic protein (GFAP) expression. The cafeteria diet reduced PV-positive neurons in the granular layer, hilus, and CA1, as well as NPY-positive neurons in the hilus, without altering other GABAergic populations or overall GABA levels. The high-sugar diet induced a decrease in the number of PV-positive cells in CA3 and an increase in CB-positive cells in the hilus and CA1. No alterations were observed in the cholinergic varicosities. The cafeteria diet also reduced the relative mRNA expression of RELN without significant changes in BDNF and CDK5 levels. The cafeteria diet increased the number but reduced the length of the astrocyte processes. These data highlight the significance of determining the mechanisms mediating the observed effects of these diets and imply that the cognitive impairments previously found might be related to both the neuroinflammation process and the reduction in PV, NPY, and RELN expression in the hippocampal formation.

## 1. Introduction

Obesity and overweight are well-known problems in contemporary society. The World Health Organization (WHO) reported that in 2016, there were over 340 million obese or overweight children and adolescents (aged 5–19) [1]. Western diets based on refined sugars, highly fatty foods, and refined grains, typical of Occidental societies, have severe consequences for general health, such as obesity, insulin resistance, and cardiovascular complications [2,3,4,5,6,7], as well as deleterious effects on the central nervous system [8]. Indeed, even short periods of Western diet consumption may compromise cognitive function [9,10], and cognitive impairments may emerge even before changes in body weight and obesity development [10,11,12,13,14]. Another determinant factor is the age at which high-caloric diets begin, particularly for the brain, since crucial events in brain development, neurogenic processes, and maturation occur during early life [15,16]. High-caloric diets affect learning and memory performance in an age-dependent manner and are more deleterious in young rats [17,18]. In the present study, we focused on analyzing the effects of high-caloric diets in the hippocampus, particularly in the gamma-aminobutyric acid (GABA)ergic system, and its relationship with the cholinergic system, neurogenesis, and neuroinflammation.

The hippocampus is one of the areas most susceptible to nutritional imbalances, with impairment of spatial learning [19,20] and memory [21,22,23] and changes in anxiety [24,25,26] after high-caloric diet treatment. In the hippocampus, a considerable proportion of the neuronal population comprises GABAergic interneurons, including neurons containing parvalbumin (PV), calretinin (CR), calbindin (CB), neuropeptide Y (NPY), and somatostatin (SST). Even though previous studies have shown the importance of the GABAergic interneuronal population in cognition and emotional processes [27], information on how high-caloric diets affect the GABAergic system is still scarce.

Given that the activity of cortical GABAergic neurons is intimately related to the cholinergic system [28,29,30] and that diet composition interferes with the cholinergic system in the hippocampus [31,32,33,34], we aimed to investigate the effects of high-caloric diets on cholinergic fibers in the hippocampus by targeting vesicular acetylcholine transporter (VAChT) varicosities.

Furthermore, it is known that diet composition could induce alterations in the neurogenic process and that high-caloric diets decrease neurogenesis in the hippocampus [11,15,18,35]. Adult neurogenesis is a complex process with several well-defined steps that include the regulation of stem cell niches, cell proliferation, differentiation and de novo formation of axons and dendrites, and finally, the integration of these new neurons in pre-existing neuronal circuits [35,36]. Recent studies have identified many key factors essential for the regulation and proliferation of neuroprogenitor cells [36]. However, less is known about the factors that control the migration and functional recruitment of adult-generated neurons, and even less about how high-calorie diets affect these processes of adult neurogenesis. In this way, we selected a series of genes encoding proteins that may be altered by the consumption of high-calorie diets, including brain-derived neurotrophic factor (BDNF) [4,37], reelin (RELN) [38,39], and cyclin-dependent kinase-5 (CDK5) [40,41].

Astrocytes are non-neuronal cells with a star-shaped morphology that are responsible for ensuring neuronal survival, formation, and maintenance of synapses, as well as for maintaining the blood-brain barrier [42]. It is known that the consumption of dietary fats and high-caloric diets increases hippocampal neuroinflammation [43]. Furthermore, recent studies have shown that consumption of high-caloric diets, even for short periods, can change the morphology and plasticity of astrocytes, impairing their function [44,45,46]. Neuroinflammation can influence the metabolic condition of the brain and change the extracellular neurotransmitter levels, which precede brain dysfunction [45]. Therefore, we investigated the effects of both high-caloric diets on astrocytes, using glial fibrillary acidic protein (GFAP), and analyzed their relationship with the GABAergic system.

Most obesity-related studies have focused on high-fat diets or high-sugar diets separately, but because Western diets have both components [47], it is important to investigate them together for comparative purposes. We started from a previous study, where we found that cafeteria diet-induced alterations in spatial learning and memory were associated with a decrease in neurogenesis [18]. In the present study, using the same animals, we aimed to better understand the possible mechanisms underlying these alterations. To this end, we compared the effects of two different high-caloric diets administered for 12 weeks, one rich in sugar (HS, 30% sucrose liquid solution) and another inspired by a cafeteria diet (CAF) rich in refined sugar and saturated fat. We focused on the GABAergic system, looking for the calcium-binding proteins (CBPs) PV, CR, and CB, as well as the neuropeptides SST and NPY in the key regions of the hippocampal formation (HF), as well as overall GABA levels, using glutamate decarboxylase 1 (GAD1), an enzyme involved in the synthesis of the majority of GABA content [48]. As the expression of neuropeptides may be dependent on the cholinergic system, we also analyzed the levels of VAChT. To better understand the alterations in neurogenesis, we analyzed the effects of these diets on the mRNA levels of BDNF, RELN, and CDK-5. Finally, we analyzed astrocyte morphology and its relationship with the GABAergic system and neurogenesis.

## 2. Results

### 2.1. Body Weight, Caloric Consumption, and Fat Mass

The body weights of the animals (g ± (SEM)), caloric consumption (Kcal ± (SEM)), and total fat mass body (% body weight (BW) ± (SEM)) across the experiment were already published in our previous work [18]. In summary, after treatment with the different diets for 12 weeks, the body weights increased to 435 g (9) in control (CT) rats, 410 g (15) in high-sugar (HS) rats, and 460 g (14) in cafeteria (CAF) rats. On average, CAF-treated rats were 47 g heavier than HS rats (*p* < 0.05), and there was a slight reduction in the weight of HS-treated rats compared to controls (not significant). Relative to caloric consumption, it was observed that the intake per cage (two rats per cage) during the whole treatment (12 weeks) was 14,453 kcal (220) for control rats, 15,223 kcal (155.8) for HS rats, and 20,120 kcal (60.8) for CAF rats. HS rats consumed more calories than controls (*p* < 0.05), and CAF rats consumed more calories than control and HS rats (*p* < 0.001). Relative to the fat mass body, the results were 4.12%BW (0.24) in controls, 6.28%BW (0.55) in HS, and 8.54%BW (0.31) in CAF animals. CAF-fed animals had more adipose tissue than the control (*p* < 0.0001) and HS-treated rats (*p* < 0.001), and HS-fed rats had more adipose tissue than the controls (*p* < 0.001). For more detailed results, please see Ferreira et al., 2018 [18].

### 2.2. Neuronal Density

#### 2.2.1. PV-Positive Neurons Areal Density in the Dentate Gyrus (DG), CA3, and CA1 Regions

The areal density estimates of PV-IR neurons in the molecular and granular layers and hilus subregions of the DG, as well as in the CA3 and CA1 regions, are shown in Figure 1. ANOVA revealed a significant effect of treatment on the areal density of PV-IR cells in the granular layer (F(2,15) = 5.74, *p* < 0.05) and hilus (F(2,15) = 11.91, *p* < 0.001) but not in the molecular layer (F(2,15) = 0.44, n.s.) of the DG. For the CA3 and CA1 regions, we found a significant effect of treatment in the areal density of PV-IR neurons in the CA3 (F(2,15) = 4.94, *p* < 0.05) and CA1 areas (F(2,15) = 9.44, *p* < 0.01). CAF treatment significantly reduced the number of PV-IR cells in the granular layer (*p* < 0.05), hilus (*p* < 0.01), and CA1 (*p* < 0.05, versus HS) compared to controls and HS-treated rats, but no differences were found between HS and controls. In the CA3 region, only the HS diet induced a reduction in the number of PV-IR neurons compared with that in control animals (*p* < 0.05).

#### 2.2.2. CR-Positive Neurons Areal Density in the DG, CA3, and CA1 Regions

The areal density estimates of CR-IR neurons in the DG, CA3, and CA1 are shown in Figure 2. ANOVA revealed that there was no significant effect of diet in the molecular layer (F(2,15) = 0.022 n.s.), granular layer (F(2,15) = 1.164, n.s.), hilar region (F(2,15) = 0.637, n.s.), CA3 region (F(2,15) = 0.223, n.s.), or CA1 region (F(2,15) = 1.734, n.s.).

#### 2.2.3. CB-Positive Neurons Areal Density in the DG, CA3, and CA1 Regions

The areal density estimates of the CB-IR neurons in the DG, CA3, and CA1 regions are shown in Figure 3. ANOVA showed that there was no significant effect of treatment on the molecular layer (F(2,15) = 0.45, n.s.), granular layer (F(2,15) = 6.95, n.s.), hilus (F(2,15) = 1.14, n.s.) and CA3 region (F(2,15) = 3.32, n.s.). Conversely, ANOVA revealed a significant effect of treatment on the number of CB-IR cells in the CA1 region (F(2,15) = 8.97, *p* < 0.05). Post hoc analysis showed that the number of CB-IR cells in the CA1 region was significantly higher in rats fed the HS diet than in the control (*p* < 0.05) and CAF-treated (*p* < 0.01) rats. No significant differences were found for the other HF regions.

#### 2.2.4. SST-Positive Neurons Areal Density in the DG, CA3, and CA1 Regions

The estimates of the number of SST-IR neurons in the DG, CA3, and CA1 regions are shown in Figure 4. ANOVA showed no significant effects of treatment on the areal density of SST-IR cells in the molecular layer (F(2,15) = 0.557, n.s.), granular layer (F(2,15) = 1.230, n.s.), hilus (F(2,15) = 0.283, n.s.), CA3 region (F(2,15) = 0.699, n.s.), and CA1 region (F(2,15) = 0.725, n.s.).

#### 2.2.5. NPY-Positive Neurons Areal Density in the DG, CA3, and CA1 Regions

Estimates of the areal densities of NPY-IR neurons in the DG, CA3, and CA1 regions are shown in Figure 5. ANOVA showed a significant effect of treatment in the hilus (F(2,15) = 11.43, *p* < 0.01), but not in the molecular (F(2,15) = 1.52, n.s.) or granular (F(2,15) = 0.42, n.s.) layers. In the hilus, post hoc analysis revealed that the CAF diet caused a significant reduction in the number of NPY-IR neurons compared to the control (*p* < 0.001) and HS-treated (*p* < 0.05) groups. No significant differences were observed between the HS-treated and the control groups. Regarding the CA3 and CA1 regions, we found no significant effects of diet in the CA3 region (F(2,15) = 0.87, n.s.) or the CA1 region (F(2,15) = 3.51, n.s.).

#### 2.2.6. VAChT-Positive Varicosities Areal Density in the Hilus

The estimates of the areal density of VAChT-IR varicosities in the hilar region of the DG are shown in Figure 6. ANOVA showed that there were no significant effects of treatment on the density of cholinergic varicosities marked by the VAChT (F(2,15) = 2.09, n.s.).

### 2.3. mRNA Relative Expressions in the HF

#### 2.3.1. *GAD1*

In Figure 7, we present the relative mRNA expression of *GAD1* in the HF. ANOVA showed that there was no significant effect of treatment (F(2,9) = 0.11, n.s.) on *GAD1* levels in the HF.

#### 2.3.2. *BDNF*

The estimates of the relative mRNA expression of *BDNF*, a gene related to cell proliferation in HF, are shown in Figure 8. ANOVA showed no significant effect of treatment on the expression of this gene (F(2,9) = 0.88, n.s.).

#### 2.3.3. *RELN*

The estimates of the relative mRNA expression of *RELN* in HF are shown in Figure 9. ANOVA showed a significant effect of treatment on *RELN* expression in the CAF group (F(2,9) = 5.16, *p* < 0.05). Post hoc tests revealed that animals fed the CAF diet had a significant decrease in *RELN* mRNA relative expression compared to control animals (*p* < 0.05). There were no significant differences between HS-treated and control rats or between HS- and CAF-treated rats.

#### 2.3.4. *CDK5*

Estimates of the relative mRNA expression of *CDK5* in HF are shown in Figure 10. ANOVA showed no significant effect of diet on the expression of *CDK5* (F(2,9) = 2.368, n.s.).

### 2.4. Astrocytes

The effects of consuming a high-caloric diet on the length distribution of astrocyte processes are shown in Figure 11 and Figure 12. ANOVA revealed a significant effect of dietary treatment on hilus (F(2,15) = 11.87, *p* = 0.0008) but not on the CA3–CA1 region (F(2,15) = 0.5939, *p* = 0.5647). Post hoc analysis revealed that the CAF-treated animals had more processes with a smaller branch length in the hilus region (*p* < 0.01, between 0.1–2.5 µm; and *p* < 0.05, between 2.5–4.5 µm and 4.5–6.5 µm) than in controls. The results also revealed that in the CA3–CA1 region, the HS-treated animals had fewer processes with a smaller branch length (between 0.1 and 2.5 µm) than the CAF-treated rats (*p* < 0.01) and controls (*p* < 0.01). For the higher length intervals, there were no differences between groups, both for the hilus and CA3–CA1 regions.

Beyond the length distributions, we examined the number of astrocytes per area, as shown in Figure 13, for the same regions. ANOVA revealed that there were no significant effects of dietary treatment on the hilus region (F(2,15) = 1.581, *p* = 0.238) or the CA3–CA1 region (F(2,15) = 2.986, *p* = 0.081). We found no difference in the number of astrocytes per unit area between the groups in the two regions.

Since we observed statistical differences in the length of astrocyte processes in both regions examined, we decided to analyze the morphology of astrocytes, as shown in Figure 14, by counting the number of processes per astrocyte. ANOVA revealed a significant effect of dietary treatment on the hilus (F(2,15) = 11.05, *p* = 0.001) but not in the CA3–CA1 region (F(2,15) = 0.756, *p* = 0.486). Post hoc analysis revealed that, in the hilus, there was a significant increase in the number of processes per astrocyte in CAF-treated animals compared to HS-treated (*p* < 0.01) and control rats (*p* < 0.01). The number of processes per astrocyte remained unchanged in the CA3–CA1 region.

## 3. Discussion

Knowing that cafeteria diets, rich in both fat and sugar, induce changes in anxiety levels and learning and memory [18], we decided to analyze potential neuronal alterations in the hippocampus, a fundamental region for these behaviors and functions. Therefore, in this study, we compared two distinct high-caloric diets, CAF and HS, that were administered to juvenile rats for 12 weeks and focused on the analysis of the GABAergic system and its relationship with the cholinergic system, neurogenesis, and astrocyte morphology.

Interestingly, when we looked at the body weight evolution during the 12-week experimental period, we realized that the increase in weight was similar in all groups (for details, please see Ferreira et al., 2018 [18]). This means that, although the high-caloric diets had more calories than the standard chow, there was no significant difference in weight gain between rats fed with these diets and controls, i.e., at the end of the treatments, there were no significant differences in weight between CAF and control or between HS and control rats. Indeed, the weight gain of CAF-treated rats was more evident than that of HS-fed rats and, at the end of the experiment, CAF-treated rats were significantly heavier than HS-fed rats. It has already been verified that high-caloric diets do not necessarily induce weight gain in juveniles [11], given their high activity. Naturally, if the treatments were extended for more weeks, we would probably find a significant difference in weight gain in both groups relative to controls, not only because of the longer time of treatment but also because the administration of the high-caloric diets would achieve the adult phase when animals are less active. It is important to note that there was an increase in body fat mass in both CAF- and HS-fed animals, which is in line with data from previous works [50,51,52] showing that the CAF and HS diets are capable of inducing a significant increase in body fat mass, even in juveniles. However, in fact, there were no significant differences in weight between the control and HS animals, at the end of the treatments, despite HS animals having significantly more adipose tissue. However, an increase in adipose tissue does also not necessarily imply an increase in weight. In fact, several studies have proven that HS diets may not cause an increase in weight [53,54,55], even with an increase in fat mass [53,54,56]. Although the HS diet does not increase weight or cause obesity, as already described before [55], we cannot ignore its potential deleterious effects and, therefore, it is also important to understand its effects on the brain and compare it with the diets rich in fat and sugar.

Despite the widespread distribution of the GABAergic system throughout the HF, the impact of high-caloric diets on interneurons is not yet completely known. Previous studies have shown that high-fat diets induce a decrease in vesicular GABA transporter [57] and GABA concentration [58] in several brain regions, including HF [57,58]. To analyze the GABA levels, we used GAD1 mRNA (encoding the GAD67 enzyme) since GAD1 accounts for 80–90% of overall brain GABA [48] and because GAD1 expression is a reliable proxy of altered GABAergic transmission [59]. However, in the present study, none of the high-caloric diets induced significant changes in GAD1 mRNA levels in the hippocampus. The discrepancy among existing studies may be explained using different diet protocols that use distinct fat/sugar ratios and/or duration of diet exposure. In this way, the 12 weeks of high-caloric diet treatments may not have been sufficient to induce changes in GABA levels in the hippocampus using our protocol. Although the present high-caloric diets induced alterations in specific GABAergic populations (as we will discuss in the following paragraphs), they do not severely impact the global GABAergic level. Moreover, we cannot rule out the hypothesis that the methodology used to measure GABA content in the hippocampus did not have enough sensitivity to detect possible differences. In other words, the high-sugar and high-sugar-high-fat diets may induce changes in GABA levels that we were unable to detect using GAD1 mRNA.

To analyze the effects of high-caloric diets on GABAergic neurons, we focused on the interneurons expressing CBPs. Interestingly, we found that the CAF diet induced a significant reduction in the density of PV-IR neurons in the hilus and granular layer of the DG as well as in the CA1 region, whereas the high-sugar diet only induced a significant decrease in the CA3 region. This reduction in PV-IR cell density may be due to cell death, a decrease in activity, or an alteration in protein content; however, to fully understand this issue, future studies using nutritional rehabilitation, aiming to estimate total cell numbers, should be performed. Our results corroborate previous studies, where it was found that high-fat–high-sugar diets induce a reduction in PV-IR neuronal expression in the prefrontal cortex [60] and CA1 region [61] of rodents. However, in our study, we found that the CAF diet caused a reduction in PV not only in CA1 but also in the hilus and granular layer. Moreover, the reduction in PV in the CA3 region of HS-treated rats is also in line with a previous study in which an HS diet fed to juvenile rats induced a reduction in PV in the DG, CA3, and CA1 regions of the HF [39]. Taken together, these results clearly show that interneurons expressing PV are particularly vulnerable to high-caloric diets, at least during the juvenile period. It is important to note that PV expression in the hippocampus is residual in the first ten postnatal days and then increases to mature levels between P12 and P30 [62]. The majority of PV-IR interneurons in the HF belong to the GABAergic perisomatic inhibitory neuronal group and are positioned for the fine-tuning and control of the principal efferent neurons of the HF [63]. Thus, our results suggest that high-caloric diets will damage the perisomatic inhibitory circuitry of the hippocampus, which will change the excitatory/inhibitory balance in this region. Consequently, the deleterious impact of high-caloric diets in the hippocampal PV-expressing interneurons of the hippocampus might account for the spatial learning and memory deficits [64,65,66] and the increased anxiety [67] levels that we previously reported [18].

Conversely, we did not find a significant effect of high-caloric diets on the density of CR-IR cells in the HF. Studies that have analyzed the effects of high-caloric diets on CR expression are scarce [68,69] and the present work is the first one to examine CR expression in the HF. While PV-IR neurons act primordially on principal neurons, CR is a special GABAergic population because they synchronize [70] and target other GABAergic cells almost exclusively, including vasoactive intestinal polypeptide-, CB-, SST-, and other CR-IR neurons, but avoid PV-IR neurons [71,72]. Thus, the present results show that none of the high-caloric diets impair the fine-tune synchronization of the inhibitory drive upon principal neurons made by CR-IR interneurons. In addition, it is clear that the high-caloric diets differentially impact the PV- and CR-IR cells of the hippocampus during the juvenile period, apparently driving the GABAergic system to a more immature state since it is known that PV-mediated plasticity in the hippocampus may continue into early adulthood.

Regarding CB, we did not find a major impact of the high-caloric diets on the density of CB-IR cells in the HF, although there was an increase in the density of CB-positive cells in the hilus that, however, it did not achieve a significant level. Moreover, we found a slight increase in CB-IR in the HS-treated rats that were confined to the CA1 region. This was not completely unexpected since there is some co-localization between CB and CR [73] and because the levels of GAD1 in HF were unchanged. Related to high-caloric diet consumption, we were not able to find any studies that have analyzed its effects on CB expression. CB acts as a transporter of intracellular Ca^2+^, promotes synaptic plasticity, and differs from other CBPs since it shows cell-specific subtype expression patterns in the hippocampus [74]. In addition, CB is known to have a role in learning and memory, and its overexpression in DG neurons disrupts spatial memory [74]. In line with this, the suggestive increased density of CB-IR neurons in the hippocampus could lead, or at least contribute, to the spatial memory impairment described in the same animals in our previous work [18].

Except for the reduction in the density of PV-IR cells, it seems that high-caloric diets do not severely impact CBPs levels. However, we cannot discard the hypothesis that a reduction in the density of PV-IR cells could lead to changes in calcium homeostasis, particularly calcium reuptake, and consequently contribute to the deleterious alterations [75] observed in high-caloric diets.

In the present study, we found that the CAF diet induced a significant decrease in the density of NPY-positive neurons in the HF, but only in the dentate hilus. Interestingly, this effect of the CAF diet seems to be specific to NPY, as the density of interneurons immunoreactive to SST, a neuropeptide known to be co-expressed by most hilar GABAergic interneurons, including those that produce NPY, did not change. This reduction in the density of NPY-immunoreactive cells may be related to several mechanisms, including cell death or reduction of expression or decreased activity, or alterations in protein content [76,77]. To clarify the underlying cause of such variation, future studies are needed, including studies with nutritional rehabilitation. Even though earlier studies have already reported a decrease in the expression of NPY in the hippocampus in rats fed a high-fat diet [77], our study is the first one to show that a CAF diet reduces neuropeptides in the hippocampus, namely the NPY-IR neuronal population and that this reduction occurs only in the dentate hilus, where the NPY-positive interneurons are more concentrated [78]. Knowing that NPY-positive cell subpopulations can also inhibit other subpopulations of interneurons [78], it is conceivable that such decrease may interfere with the activity of the other local interneuronal populations, leading or contributing to spatial learning and memory impairment in CAF-fed rats, as we have shown in a previous study [18].

Considering that the activity of NPY- and SST-ergic neuronal subpopulations could be dependent on cholinergic innervation in the cerebral cortex [29,30] and that HF interneurons are directly related to the cholinergic system since this system is involved in the recruitment of interneurons [79], we decided to analyze the effects of these high-caloric diets on the cholinergic system of the HF. Interestingly, we did not find significant changes in VAChT levels in the dentate hilus, suggesting that the incorporation of acetylcholine into synaptic vesicles in the hippocampus is not affected by the high-caloric diet treatments and that the reduction in the density of NPY-IR neurons in the dentate hilus observed in CAF-fed animals is unlikely to be due to alterations in expression of VAChT levels. These results are supported by a previous study that found that the cholinergic system was not affected by a high-fat diet or by consuming a Western diet [21]. Furthermore, the cognitive impairment that we previously found using the CAF diet [18] is probably not directly related to the cholinergic system [21]. Notably, an earlier study found a decrease in acetylcholinesterase in the hippocampus of rats after treatment with high-caloric diets [80]. However, in the aforementioned study [78], the diet was extended for six months, and the methodology used to evaluate the cholinergic system was different from that used in the present study, which could explain the discrepancy between the results.

Neurogenesis is a mechanism that likely underlies the cognitive alterations found in CAF-fed rats. To evaluate possible alterations in neurogenesis we chose BDNF because it is involved in the proliferation and/or survival of cells [35] and is a neurotrophin essential for neurite outgrowth and synaptic strengthening [81]. Moreover, BDNF regulates the expression of PV and other CBPs [82]. Interestingly, we found that none of the high-caloric diets induced significant alterations in BDNF mRNA levels in the HF. These results are at odds with those obtained using other high-caloric diets, where it was shown that in rats, there is a reduction in BDNF expression that correlated with memory deficits [83], and in mice, there is both an increased BDNF expression [25] in the hippocampus and a decreased expression [84]. Indeed, our results are in agreement with other studies in which high-caloric diets did not induce alterations in BDNF levels [14,85,86], even in the presence of cognitive impairment [14,39,87]. However, based on the work of Molteni et al. [81], it is plausible to assume that feeding a high-caloric diet for longer periods might affect the levels of BDNF. Nevertheless, the present results show that the spatial learning and memory alterations and the decreased doublecortin (DCX) levels previously reported [18] are probably not directly related to alterations in BDNF levels in the HF.

The present study is a pioneer in showing that both CAF and HS diets do not cause significant alterations in the levels of CDK5 in HF. It is known, however, from other studies, that CDK5 ablation in hippocampal neural progenitor cells leads to a decrease in DCX-IR neurons [88] that CDK5 acts upon DCX during adult neurogenesis, and that when CDK5 is inactivated, it results in aberrant formation of newborn cells [41]. Taken together, it seems that the reduction in DCX that we previously observed in CAF-fed rats [18] can be mainly related to alterations in the proliferation or migration of progenitor cells and not to changes in their survival or maturation, although we cannot exclude this possibility because other factors, that we did not analyze, could be involved.

In previously reported data [39], we found that the mRNA expression levels of RELN were significantly decreased in the hippocampus of rats fed the CAF diet. RELN is an extracellular matrix protein that is crucial for neuronal migration during the development of different brain regions [89,90] and is expressed during adulthood in hippocampal and cortical interneurons [90,91]. RELN is also known to be preferentially synthesized by GABAergic neurons in the adult rodent brain [91,92]. This inactivation of the RELN signaling pathway, specifically in adult neuroprogenitor cells, induces aberrant migration, formation of ectopic dendrites in the dentate hilus, establishment of aberrant circuits, and decreased dendrite development [90]. Furthermore, it was previously shown that in the adult brain, RELN regulates neurogenesis and migration, as well as the structural and functional properties of synapses [93]. Thus, the finding that the CAF diet leads to a reduction in DCX expression [18] may be related to a reduction in reelin levels in the hippocampus. Some studies demonstrate that reelin is an important factor in the regulation of neurogenesis in adults, regulating dendritic migration and development [90,93]. Furthermore, some works demonstrate that adult neurogenesis is fundamental for spatial learning and memory [94,95]. For this reason, we suggest that the changes in learning and memory observed in animals treated with a cafeteria diet [18] may be related to the decrease in the expression of DCX and reelin. Finally, because RELN is also fundamental for the modulation of the structural and functional plastic properties of adult synapses, including the induction and maintenance of long-term potentiation (LTP) [93], it is plausible to assume that the learning and memory impairment detected in CAF-treated rats [18] could be attributed to potential alterations in the HF synapses consequent to the reduction in RELN levels.

As mentioned before, impairment in the expression of CBPs in specific regions of the hippocampus could, ultimately, lead to an increase in circulating Ca^2+^ levels since it was not taken up [75]. In addition, non-neuronal cells have the capacity to uptake calcium [96]. Regarding the non-neuronal populations, astrocytes are glial cells that interact with neurons and ensure that they receive and propagate information [42]. However, after brain insults, such as the consumption of high-caloric diets, astrocytes can become activated, shifting their function, and leading to an inflammatory process known as neuroinflammation [5]. The consumption of high-caloric diets [46] and the uptake of circulating Ca^2+^ [97] have been shown to induce morphological alterations in astrocytes. Actually, the majority of astrocytic regulatory functions rely on their processes [45]. In addition, our results show that the CAF diet alters the astrocyte morphology through the increased number of processes in the hilus region, as has already been described in other studies [46,98,99]. Indeed, we found that the astrocytes of CAF-treated rats had more processes but their lengths were reduced when compared to controls and HS-treated rats. These results are not in line with classical astrocytosis morphology, where the projections become longer and less abundant [100]. However, other studies have shown that the astrocytic processes undergo plastic changes according to nutritional states in the hypothalamus [44,45,101]. Our study is the first to show the same alteration in the hippocampus of young animals, as we had already described for high-fat-high-sugar diets in old animals [46]. It is important to note that the altered morphology of astrocytes is region-dependent, only occurring in the same regions where we detected the decreased activity of CBPs, as was the case of the hilus in CAF-fed rats, where we observed a decreased density of PV-IR neurons and altered morphology of astrocyte processes.

## 4. Methods and Materials

### 4.1. Animals and Diets

Thirty male Wistar rats bred at the animal facility of the Faculty of Medicine of the University of Porto, Portugal, were kept under standard laboratory conditions (20–22 °C and a 12 h light/dark cycle) with food and water ad libitum. Rats were housed twice per cage to avoid social isolation and to allow for the daily quantification of liquid and food consumption. They were weighed weekly, and the bedding was changed simultaneously to minimize the stress caused by handling. At 4 weeks of age, the rats were randomly assigned to one of the following three groups: the control group (CT, n = 10) had free access to tap water and was fed standard rat chow (Mucedola, 4RF1, Italy) containing proteins (17%), lysine (0.7%), methionine (0.3%), cysteine (0.5%), carbohydrates (57%), fat (4%), and salts (7%) (Table 1). The standard chow provided approximately 3.9 Kcal/g, 20% of energy as protein, 12% as fat, and 68% as carbohydrates; the high-sugar (HS)-treated group (HS, n = 10) drank a solution of 30% sucrose (Sigma-Aldrich Company Ltd., Madrid, Spain; 1.2 Kcal/mL) instead of water, and were fed with the same standard laboratory chow (Table 1); the cafeteria (CAF)-treated rats (CAF, n = 10) drank a solution of 15% sucrose (Sigma; 0.6 Kcal/mL) and were fed with assorted food composed of standard rat chow, chocolate cake, biscuits, dog roll and a high-fat rat chow (40% fat), providing an average of 4.5 Kcal/g, approximately 12% of energy as protein, 45% as fat and 43% as carbohydrates, in addition to that provided by the standard chow and the sucrose solution (Table 1). All diets were supplemented with a vitamin fortification mixture (MP Biomedicals, Santa Ana, CA, USA). Food and liquids were available ad libitum throughout the experimental period (12 weeks) and replaced daily. The handling and care of the animals followed the guidelines of the European Communities Council Directives of 22 September 2010 (2010/63/EU) and Portuguese Act nº113/13, and was approved by ORBEA, the internal committee of the Faculty of Medicine of the University of Porto, Portugal. Possible alternatives of refinement, reduction, and replacement were all considered in the present study, and as such, efforts were made to minimize the number of animals used and their suffering.

After receiving the diets for 12 weeks, for the immunohistochemical studies, six animals from each group were randomly selected, anesthetized with sevoflurane (SevoFlo, Abbott Laboratories Ltd., Maidenhead, UK), and transcardially perfused with 150 mL of 0.1 M phosphate buffer (PB), followed by fixation with 4% paraformaldehyde in PB at pH 7.6. For mRNA studies, the remaining animals in each group (n = 4) were decapitated, and the brains were quickly removed in a cold base and stored at −80 °C until further use.

### 4.2. Immunohistochemistry

For immunohistochemistry, the brains were collected, coded for blind processing, and separated by a median cut into the right and left hemispheres. The frontal and occipital poles were removed, and the blocks containing the HF were separated and processed. As there is evidence that the HF of rodents displays right/left asymmetries [102], blocks were alternately sampled from the right and left hemispheres. The same blocks were immersed for 1 h in the same fixative used for perfusion and maintained overnight in a solution of 10% sucrose in PB at 4 °C. Using a vibratome, the brains were serially sectioned in the coronal plane at 40 μm through the HF, and sections were collected in phosphate-buffered saline (PBS). Sections of each animal were selected using a systematic, random sampling procedure in a fraction of 1:12 for each of the following sets: PV, CR, CB, NPY, SST, VAChT, and GFAP. Selected sections were washed twice in PBS and treated with 3% H_2_O_2_ solution for 7 min to inactivate endogenous peroxidase. To increase tissue penetration, 0.5% Triton X-100 was added to PBS for all immunoreactions and washing steps. The selected sections were then incubated at 4 °C for 72 h with primary antibodies against PV (Swant, Burgdorf, Switzerland, 1:5000 dilution in PBS 0.5% T), CR (Swant, 1:5000 dilution in PBS 0.5% T), CB (Swant, 1:5000 dilution in PBS 0.5% T), VAChT (Millipore, MA, USA, 1:2000 dilution in PBS 0.5% T), and GFAP (Dako, CA, USA, 1:4000 dilution in PBS 0.5% T), or overnight at 4 °C for SST (Peninsula Laboratories, Augst, Switzerland, 1:10,000 dilution in PBS 0.5% T), and NPY (Peninsula Laboratories, 1:10,000 dilution in PBS 0.5% T). The sections were then washed thrice in PBS containing 0.5%T and incubated with the respective biotinylated secondary antibodies. Afterward, the sections were treated with an avidin-biotin-peroxidase complex (Vectastain Elite ABC kit, Vector Laboratories, Newark, CA, USA, 1:800 dilution in PBS 0.5% T). The last two incubations were conducted at room temperature for 1 h. After treatment with the peroxidase complex, sections were incubated for 10 min in 0.05% diaminobenzidine (DAB, Sigma-Aldrich, Wien, Austria), to which 30 µL of 0.01% H_2_O_2_ solution was added. The sections were rinsed with PBS for at least 15 min between steps. The specificity of the immune reactions was controlled by omitting the incubation stage with primary antiserum. All immunohistochemical reactions and washings were performed in 12-well tissue culture plates to ensure that staining of sections from all groups was performed in parallel and under identical conditions. The sections were mounted on gelatin-coated slides, air-dried, dehydrated, and coverslipped using Histomount (National Diagnostics, Atlanta, GA, USA).

### 4.3. Morphometric Analysis

#### 4.3.1. Estimation of the Areal Density of PV-, CR-, CB-, NPY- and SST-Immunoreactive Cells

Immunostained brain sections were analyzed and drawn using a light microscope equipped with a camera lucida at a final magnification of 160×. The layer boundaries of the DG, CA3, and CA1 regions of the HF were consistently defined at all levels along the septotemporal axis of the HF, based on cytoarchitectonic criteria [103] and using a rat brain atlas [104]. Neurons belonging to the CA2 hippocampal field were included in the CA3 region. Immunoreactive neurons were identified as darkly stained perikarya. The number of neurons in each layer of the DG or pyramidal strata of the hippocampal CA1 and CA3 subfields was counted from the drawings. The estimation for each cell type was obtained from an average of 12 immunostained sections per rat, which were sampled as previously described. The same camera lucida drawings were used to calculate the areas of the layers. A transparent sheet bearing a test system consisting of a set of regularly spaced points was superimposed on the drawings, and the number of points falling within the boundaries of the molecular layers (MLs), granular layer (GL), hilus, CA3, and CA1 were counted. The area of each layer was estimated by multiplying the number of points within the limits by the value of the area per point of the test system (0.0096 mm^2^). The cell numbers obtained were divided by the values of the corresponding laminar areas to yield the areal density values (number of cells/mm^2^).

#### 4.3.2. Estimation of the Areal Density of VAChT-Positive Varicosities in the Dentate Hilus

Cholinergic varicosities stained with VAChT were counted using a computer-assisted image analyzer (Leica Qwin, v3.3.1) connected to a Leica DMR microscope and a Leica DC 300 F video camera (Leica Microsystems, Wetzlar GmbH, Wetzlar, Germany). For each animal, an average of 12 VAChT-stained sections was used and assessed as previously described [30]. The measurements were performed at a final magnification of 1000×. The varicosities were defined as darkly stained axonal dilations with a size greater than 0.25 μm^2^ [105]. A sample frame (3.86  ×  10^3^ μm^2^) was laid over each field of view, and the number of varicosities falling within it was counted. To obtain the mean count of the dentate hilus, four different frame positions within each region were used, each at a randomly selected position. The results are presented as area densities (number/mm^2^).

### 4.4. Quantification of GFAP-Positive Astrocytes

Immunostained brain sections were photographed using a light microscope (Zeiss Scope A.1, Zeiss, Jena, Germany) equipped with an AxioCam MRc5 (Zeiss) camera at a final magnification of 20×. The layer boundaries of the DG, CA3, and CA1 were consistently defined at all levels along the septotemporal axis of the HF based on cytoarchitectonic criteria [103] and using a rat brain atlas [104]. Astrocytes from the CA2 hippocampal area were included in the CA3 region. The images obtained from the AxioVision Rel 4.8 (Zeiss) program were used in the ImageJ (1.50i, National Institute of Health, Bethesda, MD, USA) program and ran on a macro [106] modified to count astrocytes [49] to assess the number of GFAP-immunoreactive astrocytes and the length and number of their processes.

### 4.5. RNA Extraction, Reverse Transcription, and Quantitative Real-Time Polymerase Chain Reaction (RT-qPCR)

After dissecting and homogenizing the HF from one of the brain hemispheres, total RNA was extracted using the NZYOL reagent (NZYTech, Lisbon, Portugal), followed by chloroform extraction and isopropanol precipitation. Total RNA was quantified using a NanoDrop 2000 instrument (Thermo Scientific, Fisher Scientific, Oeiras, Portugal), and the quality was controlled using a 2100 Bioanalyzer Instrument (Agilent, Santa Clara, CA, USA). Before reverse transcription, total RNA was DNase-treated with RQ1 DNase (Promega, Fitchburg, WI, USA) to remove contaminating genomic DNA. Reverse transcription was performed using the NZY First-strand cDNA Synthesis Kit (combined oligo-dT and random hexamers) (NZYTech, Lisbon, Portugal). Quantitative real-time PCR (RT-qPCR) was performed in a StepOnePlus qPCR system (Applied Biosystems, Woburn, MA, USA) using the SensiFAST SYBR Hi-Rox Kit (Bioline, London, UK) and the standard curve method. The samples were assayed in triplicate. The qPCR reaction efficiencies for all the primer sets ranged from 92% to 100%. Gene expression was normalized to the expression levels of two endogenous housekeeping genes: glyceraldehyde 3-phosphate dehydrogenase (*GAPDH*) and *β-actin*. Table 2 lists the gene-specific primers used.

### 4.6. Statistical Analysis

Body weight, caloric consumption, and fat mass data are presented as the mean ± SEM. Morphological and mRNA expression data are presented as mean ± SD. Statistical analyses and graphics were performed using GraphPad Prism (GraphPad Software v8.0.2., Boston, CA, USA). One-way ANOVA was used to analyze body weight, caloric consumption, fat mass, areal densities, relative expression of mRNAs, and GFAP expression in astrocytes, using treatment as the independent variable. Whenever appropriate, analysis of variance (ANOVAs) was performed, followed by Tukey’s HSD post hoc comparisons. Differences were considered statistically significant at *p* < 0.05.

## 5. Conclusions

Our results show that the CAF diet administered to juveniles affects the GABAergic system of the HF more severely than the HS diet. We found that the CAF diet reduces the density of PV- and NPY-positive neurons without significantly changing the majority of the GABAergic population and overall GABA levels. Furthermore, whereas BDNF and CDK5 expression remained unchanged, the CAF diet led to a significant decrease in RELN mRNA levels and induced morphological shortening of astrocyte processes in HF. In addition, the CAF diet also induces morphological shortening of astrocyte processes in HF. Taken together, we hypothesize that the reduction in neurons expressing PV and NPY in the HF, induced by the consumption of CAF diets, may induce overexcitation and change the excitatory/inhibitory balance, leading to alterations in neurogenesis and astrocyte morphology. These findings also suggest that altered astrocytes and a reduction in the levels of PV, NPY, and RELN in the HF may contribute to the cognitive impairments observed in juveniles who consume cafeteria diets. These data also consolidate evidence that early life is an extremely vulnerable period to dietary challenges and emphasize the importance of identifying the subtle molecular mechanisms that mediate the effects of diets rich in saturated fats and refined sugar on the GABAergic system, neurogenesis, and astrocyte morphology in the maturing brain.

## Figures and Tables

**Figure 1 ijms-25-05524-f001:**
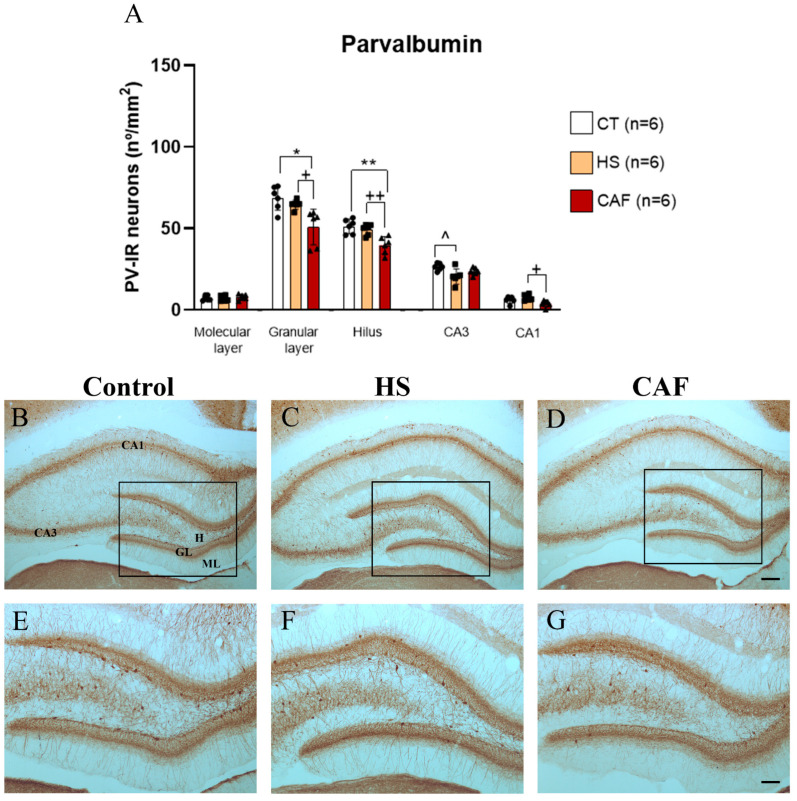
Histogram (**A**) showing the mean ± SD areal density of PV-immunoreactive (PV-IR) cells in the molecular layer, granular layer, and hilus of the DG, CA3, and CA1 regions, with 6 animals per group. The circles represent the value for each animal in the control group, squares represent the value for each animal in the HS-treated group, and triangles represent the value for each animal in the CAF-treated group. Note that CAF-treated animals showed a significant reduction in the number of PV-IR cells in the granular layer and hilus of the DG when compared to control and HS-treated rats. HS treatment induced a significant reduction in the areal density of PV-IR cells in the CA3 region compared with that in the controls. In the CA1 region, the areal density of PV-IR cells was significantly reduced in CAF-treated rats relative to that of HS-treated rats. * *p* < 0.05 and ** *p* < 0.01, CAF-treated rats versus controls; ^+^ *p* < 0.05 and ^++^ *p* < 0.01, CAF-treated rats versus HS-treated rats; ^ *p*< 0.05, HS-treated rats vs. controls. CT, control; HS, high-sugar; CAF, cafeteria. Representative photomicrographs of coronal sections through the HF of control (**B**,**E**), HS- (**C**,**F**), and CAF-treated (**D**,**G**) rats immunostained for PV. The boxes drawn in (**B**), (**C**), and (**D**) approximately delineate the regions of the DG area shown at higher magnification in (**E**), (**F**), and (**G**), respectively. High-power photomicrographs of the DG of control (**E**), HS- (**F**), and CAF-treated (**G**) rats. ML, molecular layer; GL, granule cell layer; H, dentate hilus; CA3, pyramidal cell layer of CA3 hippocampal field; and CA1, pyramidal cell layer of CA1 hippocampal field. Scale bar: 200 µm in (**B**–**D**) and 100 µm (**E**–**G**).

**Figure 2 ijms-25-05524-f002:**
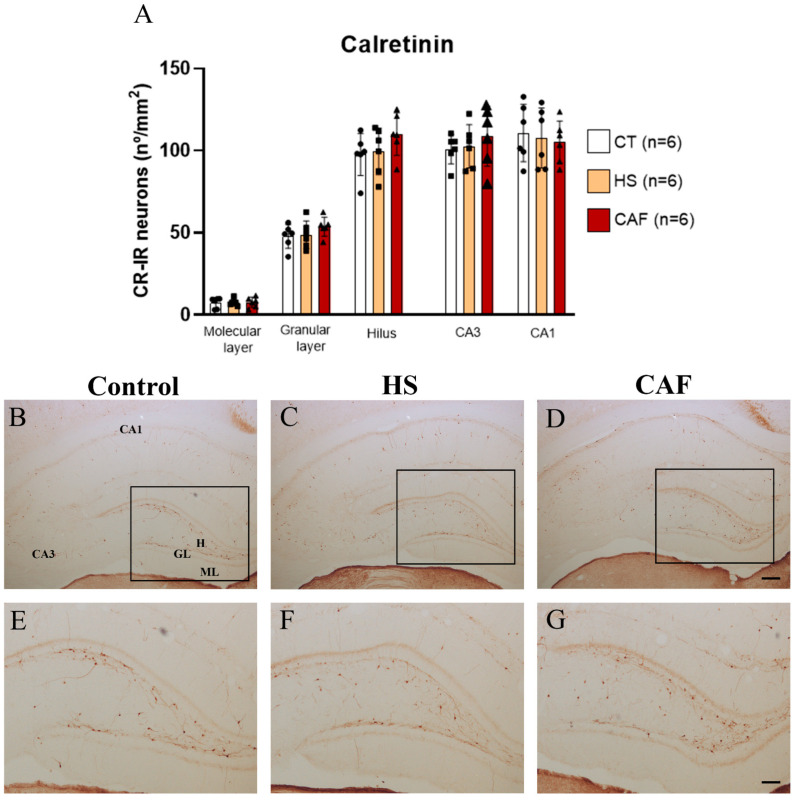
Histogram (**A**) showing the mean ± SD areal density of CR-IR cells in the DG, CA3, and CA1 regions, with 6 animals per group. Circles represent the value for each animal in the control group, squares represent the value for each animal in the HS-treated group, and triangles represent the value for each animal in the CAF-treated group. No significant effects of diet were observed in any region. CT, Control; HS, High-Sugar; CAF, Cafeteria. Representative photomicrographs of coronal sections through the HF of control (**B**,**E**), HS- (**C**,**F**), and CAF-treated (**D**,**G**) rats immunostained for CR. The boxes drawn in (**B**), (**C**), and (**D**) approximately delineate the regions of the DG area shown at higher magnification in (**E**), (**F**), and (**G**), respectively. High-power photomicrographs of DG of control (**E**), HS- (**F**), and SCAF-treated (**G**) rats. ML, molecular layer; GL, granule cell layer; H, dentate hilus; CA3, pyramidal cell layer of CA3 hippocampal field; and CA1, pyramidal cell layer of CA1 hippocampal field. Scale bar: 200 µm in (**B**–**D**) and 100 µm (**E**–**G**).

**Figure 3 ijms-25-05524-f003:**
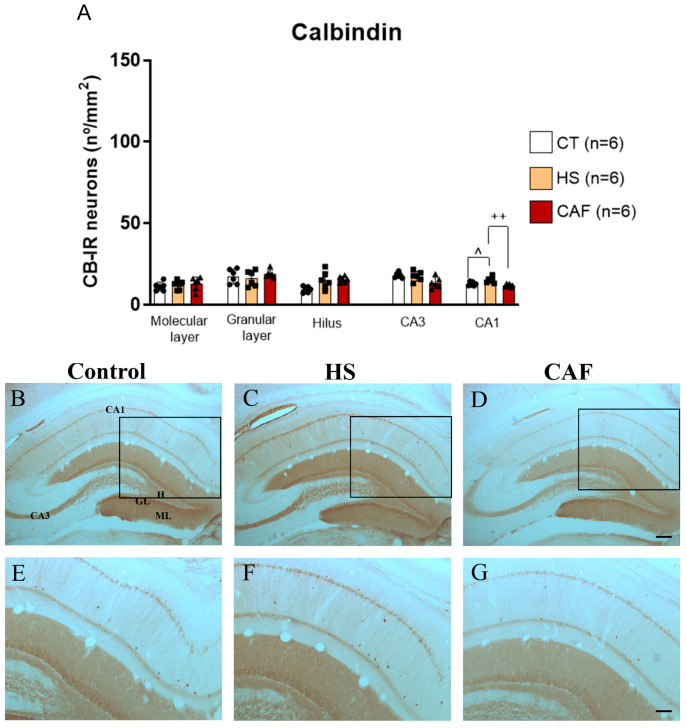
Histogram (**A**) showing the mean ± SD areal density of CB-IR cells in the DG, CA3, and CA1, with 6 animals per group. Circles represent the value for each animal in the control group, squares represent the value for each animal in the HS-treated group, and triangles represent the value for each animal in the CAF-treated group. The HS diet induced a significant increase in the areal density of CB-IR cells in the CA1 region compared to control and CAF-treated rats. ^ *p* < 0.05, HS-treated versus control rats; ^++^ *p* < 0.01, HS-treated versus CAF-treated rats. CT, control; HS, high-sugar; CAF, cafeteria. Representative photomicrographs of coronal sections through the HF of control (**B**,**E**), HS- (**C**,**F**), and CAF-treated (**D**,**G**) rats immunostained for CB. The boxes drawn in (**B**), (**C**), and (**D**) delineate approximately the regions of the CA1 area shown at higher magnification in (**E**), (**F**), and (**G**), respectively. High-power photomicrographs of the CA1 of control (**E**), HS- (**F**), and CAF-treated (**G**) rats. ML, molecular layer; GL, granule cell layer; H, dentate hilus; CA3, pyramidal cell layer of CA3 hippocampal field; and CA1, pyramidal cell layer of CA1 hippocampal field. Scale bar: 200 µm in (**B**–**D**) and 100 µm (**E**–**G**).

**Figure 4 ijms-25-05524-f004:**
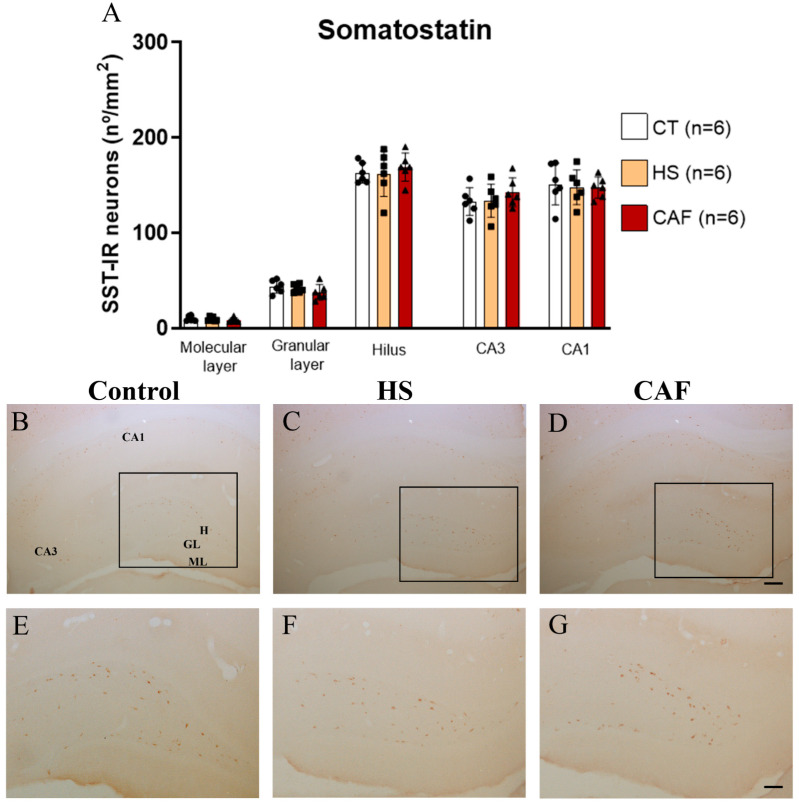
Histogram (**A**) showing the mean ± SD areal density of SST-IR cells in the DG, CA3, and CA1 regions, with 6 animals per group. The circles represent the value for each animal in the control group, squares represent the value for each animal in the HS-treated group, and triangles represent the value for each animal in the CAF-treated group. No significant changes in the number of SST-IR cells were observed. CT, control; HS, high-sugar; CAF, cafeteria. Representative photomicrographs of coronal sections through the HF of control (**B**,**E**), HS- (**C**,**F**), and CAF-treated (**D**,**G**) rats immunostained for SST. The boxes drawn in (**B**), (**C**), and (**D**) delineate approximately the regions of the DG area shown at higher magnification in (**E**), (**F**), and (**G**), respectively. High-power photomicrographs of the DG of control (**E**), HS- (**F**), and CAF-treated (**G**) rats. ML, molecular layer; GL, granule cell layer; H, dentate hilus; CA3, pyramidal cell layer of CA3 hippocampal field; and CA1, pyramidal cell layer of CA1 hippocampal field. Scale bar: 200 µm in (**B**–**D**) and 100 µm (**E**–**G**).

**Figure 5 ijms-25-05524-f005:**
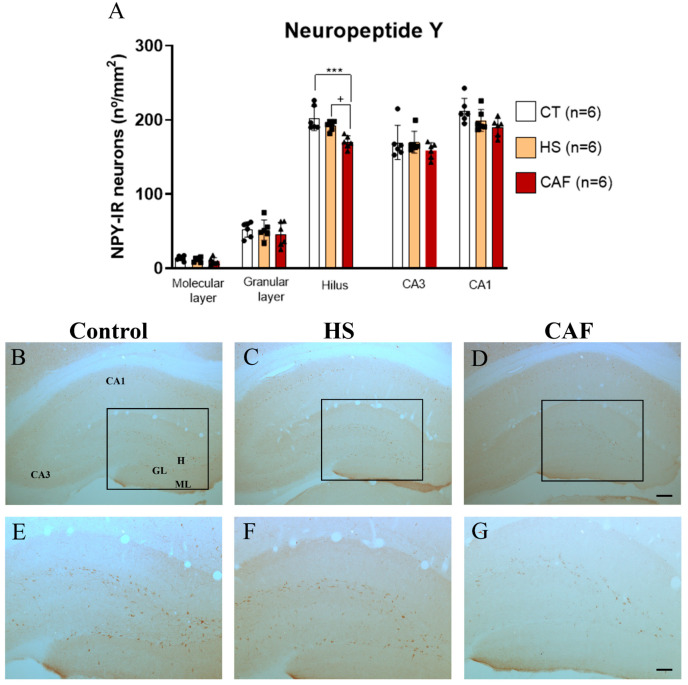
Histogram (**A**) showing the mean ± SD areal density of NPY-IR cells in the DG, CA3, and CA1 regions, with 6 animals per group. The circles represent the value for each animal in the control group, squares represent the value for each animal in the HS-treated group, and triangles represent the value for each animal in the CAF-treated group. There was a significant reduction in the number of NPY-IR cells in the hilus of CAF-treated rats compared to HS-treated and control rats. There were no significant effects of diet on any of the other HF regions. *** *p* < 0.001, CAF-treated rats versus control rats; ^+^ *p* < 0.05, CAF-treated rats versus HS-treated rats. CT, control; HS, high-sugar; CAF, cafeteria. Representative photomicrographs of coronal sections through the HF of control (**B**,**E**), HS- (**C**,**F**), and CAF-treated (**D**,**G**) rats immunostained for NPY. The boxes drawn in (**B**), (**C**), and (**D**) delineate approximately the regions of the DG area shown at higher magnification in (**E**), (**F**), and (**G**), respectively. High-power photomicrographs of the DG of control (**E**), HS- (**F**), and CAF-treated (**G**) rats. ML, molecular layer; GL, granule cell layer; H, dentate hilus; CA3, pyramidal cell layer of CA3 hippocampal field; and CA1, pyramidal cell layer of CA1 hippocampal field. Scale bar: 200 µm in (**B**–**D**) and 100 µm (**E**–**G**).

**Figure 6 ijms-25-05524-f006:**
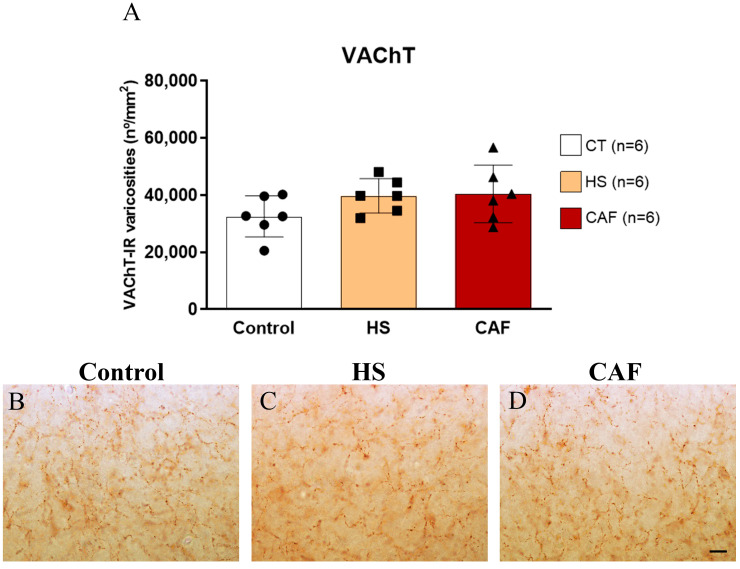
Histogram (**A**) showing the mean ± SD areal density of VAChT-IR varicosities in the hilar region of the HF with 6 animals per group. The circles represent the value for each animal in the control group, squares represent the value for each animal in the HS group, and triangles represent the value for each animal in the CAF group. There were no significant differences between groups in the density of the VAChT-IR varicosities. CT, control; HS, high-sugar; CAF, cafeteria. Representative photomicrographs of level-matched coronal sections of the dentate hilus from control (**B**), HS- (**C**), and CAF-treated (**D**) rats immunostained for VAChT. Scale bar: 10 µm in (**B**–**D**).

**Figure 7 ijms-25-05524-f007:**
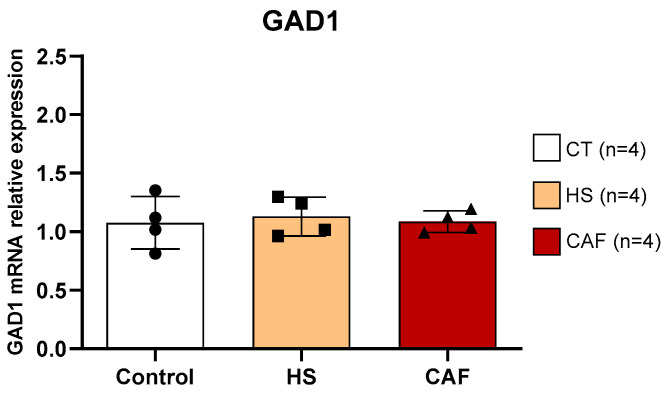
Hippocampal *GAD1* mRNA expression relative to the mean of the housekeeping gene *GAPDH/β-actin*. Data are expressed as the mean ± SD, with 4 animals per group. The circles represent the value for each animal in the control group, squares represent the value for each animal in the HS-treated group, and triangles represent the value for each animal in the CAF-treated group. CT, control; HS, high-sugar; CAF, cafeteria.

**Figure 8 ijms-25-05524-f008:**
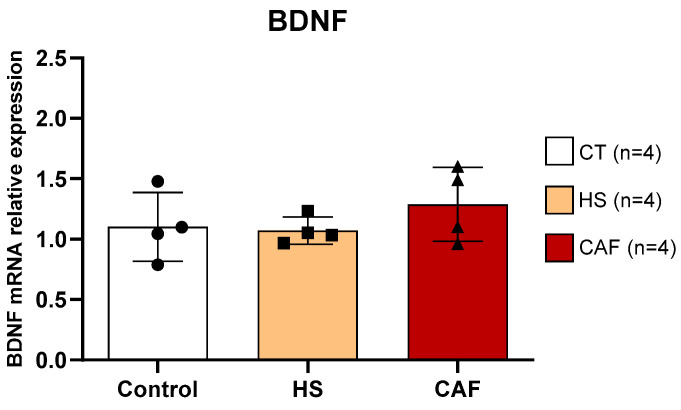
Hippocampal *BDNF* mRNA expression relative to the mean of the housekeeping gene *GAPDH/β-actin*. Data are expressed as the mean ± SD, with 4 animals per group. The circles represent the value for each animal in the control group, squares represent the value for each animal in the HS-treated group, and triangles represent the value for each animal in the CAF-treated group. CT, control; HS, high-sugar; CAF, cafeteria.

**Figure 9 ijms-25-05524-f009:**
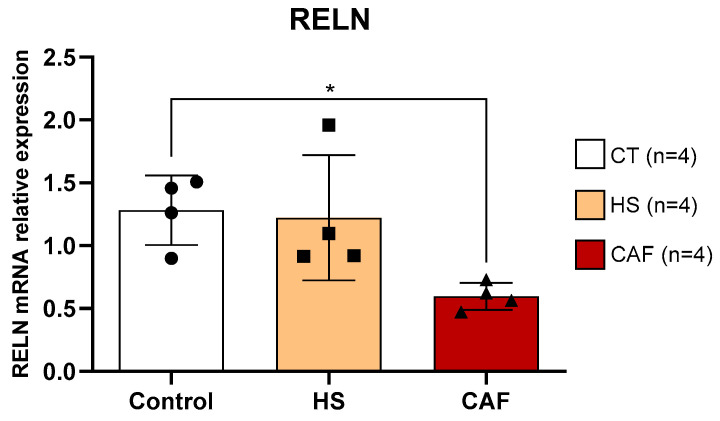
Hippocampal *RELN* mRNA expression relative to the mean of the housekeeping genes *GAPDH/β-actin*. Data are expressed as the mean ± SD, with 4 animals per group. The circles represent the value for each animal in the control group, squares represent the value for each animal in the HS-treated group, and triangles represent the value for each animal in the CAF-treated group. There was a significant reduction in RELN mRNA levels in CAF-treated rats compared to control animals. * *p* < 0.05, CAF-treated rats versus controls. CT, control; HS, high-sugar; CAF, cafeteria.

**Figure 10 ijms-25-05524-f010:**
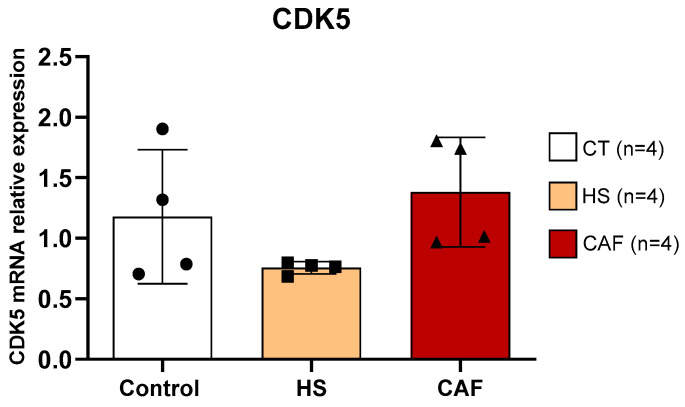
Hippocampal *CDK5* mRNA expression relative to the mean of the housekeeping genes *GAPDH/β-actin*. Data are expressed as the mean ± SD, with 4 animals per group. The circles represent the value for each animal in the control group, squares represent the value for each animal in the HS-treated group, and triangles represent the value for each animal in the CAF-treated group. CT, control; HS, high-sugar; CAF, cafeteria.

**Figure 11 ijms-25-05524-f011:**
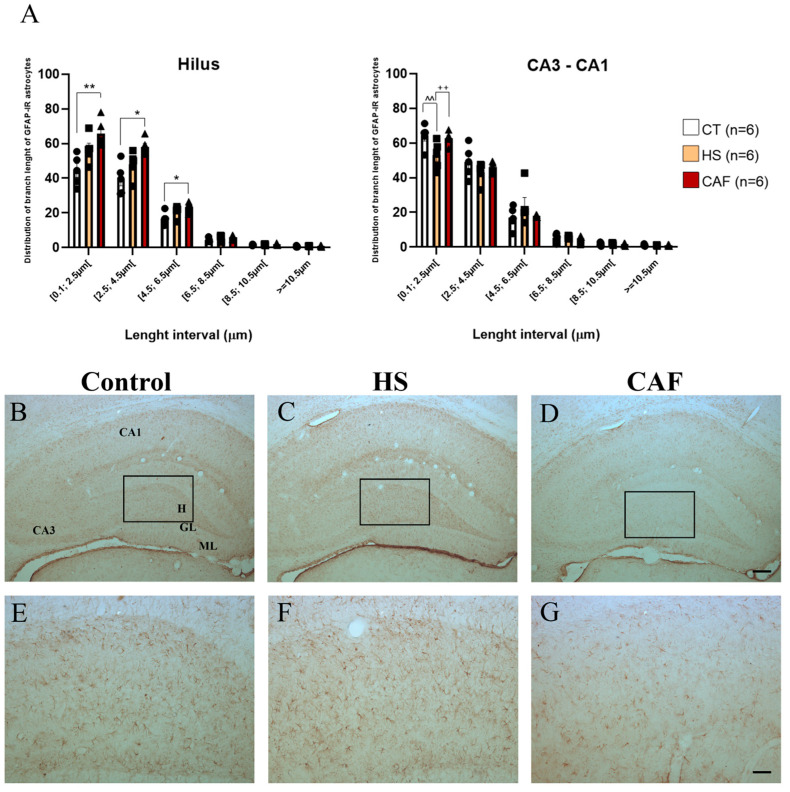
Histograms (**A**) showing the mean ± SD length distribution of astrocyte processes in the two different regions of the hippocampus. The circles represent the value for each animal in the control group, squares represent the value for each animal in the HS-treated group, and triangles represent the value for each animal in the CAF-treated group. In the hilus region, CAF-treated rats showed a significant increase in processes with a minor length distribution compared to the HS-treated and control animals. In the CA3–CA1 region, HS-treated rats displayed a significant reduction in processes with a minor length distribution when compared to controls and CAF-treated animals. * *p* < 0.05; ** *p* < 0.01, CAF-treated rats versus controls; ++ *p* < 0.01, HS-treated versus CAF-treated rats; ^^ *p* < 0.01 HS-treated vs. controls rat. CT, control; HS, high-sugar; CAF, cafeteria. Representative photomicrographs of coronal sections through the HF of control (**B**,**E**), HS- (**C**,**F**), and CAF-treated (**D**,**G**) rats immunostained for GFAP. The boxes drawn in (**B**), (**C**), and (**D**) approximately delineate the regions of the DG area shown at higher magnification in (**E**), (**F**), and (**G**), respectively. High-power photomicrographs of the DG of control (**E**), HS- (**F**), and CAF-treated (**G**) rats. ML, molecular layer; GL, granule cell layer; H, dentate hilus; CA3, pyramidal cell layer of CA3 hippocampal field; and CA1, pyramidal cell layer of CA1 hippocampal field. Scale bar: 200 µm in (**B**–**D**) and 100 µm (**E**–**G**).

**Figure 12 ijms-25-05524-f012:**
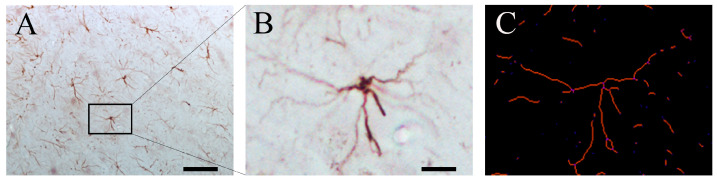
Representative GFAP-IR photomicrographs from the hippocampus (**A**) with a detailed inset of a single astrocyte (**B**) and the skeletonized detail of the inset (**C**). The black square highlights the pointed single astrocyte used in the skeletonized version to achieve the number and length of the branches, as fully detailed in [49]. Scale bar = 50 µm (**A**) and 10 µm (**B**).

**Figure 13 ijms-25-05524-f013:**
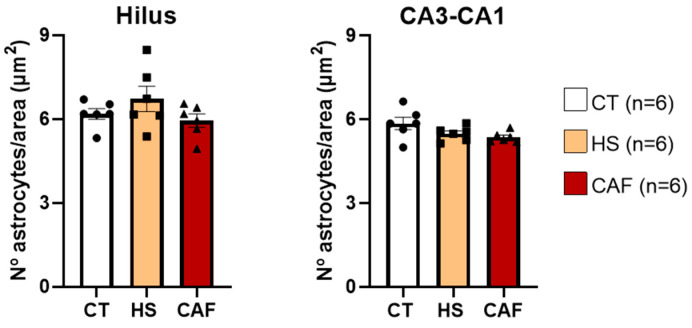
Histograms showing the mean ± SD number of astrocytes per area in the hilus and CA3-CA1 region. The circles represent the value for each animal in the control group, squares represent the value for each animal in the HS group, and triangles represent the value for each animal in the CAF group. There were no significant differences between the two regions. CT, control; HS, high-sugar; CAF, cafeteria.

**Figure 14 ijms-25-05524-f014:**
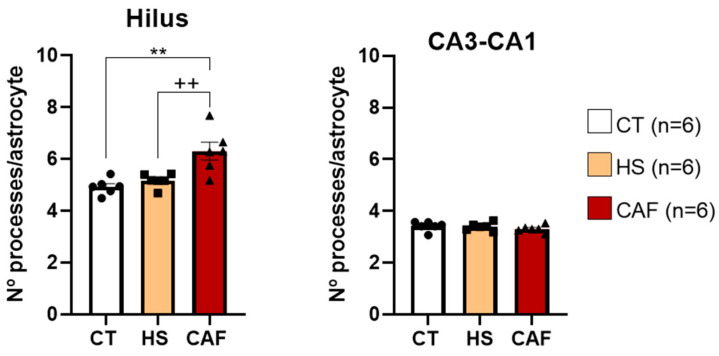
Histograms showing the mean ± SD number of processes per astrocyte in the hilus and CA3–CA1 regions of the HF. The circles represent the value for each animal in the control group, squares represent the value for each animal in the HS group, and triangles represent the value for each animal in the CAF group. There was a significant increase in the number of processes per astrocyte in the hilus region of CAF-treated animals compared to the control and HS-treated animals. ** *p* < 0.01, CAF-treated versus controls; ^++^ *p* < 0.01, CAF- versus HS-treated animals. CT, control; HS, high-sugar; CAF, cafeteria.

**Table 1 ijms-25-05524-t001:** Composition of the experimental diets.

Diet	Composition	Mucedola 4RF1	High-Sugar	Cafeteria
**Chow (%/100 g)**	Proteins	20	20	12
	Carbohydrates	68	68	45
	Fats	12	12	43
**Liquid Solution (%/100 g)**	Sucrose	0	30	15
**Total Energy (Kcal/100 g)**		390	510	510

**Table 2 ijms-25-05524-t002:** Specific forward and reverse primers for the studied genes are presented. Additionally, we used the annealing temperature for each primer pair.

Gene	Forward	Reverse	Annealing Temperature
** *GAD1* **	CCTAAAGTACGGGGTTCGCA	CAGCCATTCGCCAGCTAAAC	60 °C
** *BDNF* **	GGCCCAACGAAGAAAACCAT	TTCCTCCAGCAGAAAGAGCA	60 °C
** *RELN* **	TCAAAGACGCCTTAGCCCAG	TTCAGCGAGGTGCGAGTAAG	60 °C
** *CDK5* **	GTGACCTGGACCCTGAGATTG	ACGTTACGGCTGTGACAGAA	57 °C

## Data Availability

Data is contained within the article.

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
