# Peer review of "High-Caloric Diets in Adolescence Impair Specific GABAergic Subpopulations, Neurogenesis, and Alter Astrocyte Morphology"

_ijms, 2024, doi:10.3390/ijms25105524_

Round 1
Reviewer 1 Report (Previous Reviewer 2)
Comments and Suggestions for Authors
This revised manuscript is a significant improvement from the original version. The higher magnification images now allow readers to better appreciate differences in cell numbers between some of the different diet conditions. And the larger graphs are now more legible. The clarification that the authors are using the same mice from their 2018 behavior study does resolve some of my issues with relating these cell changes to behavioral deficits. However, this does raise a more significant problem about replicating previously published data without properly citing it.
Major point
1. The revised manuscript contains a significant ethical issue about data replication and/or self-plagiarism. The graph displaying body weight in Figure 1 in this manuscript is an exact replication of the data/graph from Figure 1 of their previous manuscript (Ferreira…Cardoso, Neurobiol Learn Mem 2018). Replicating these data/graph without properly citing it and/or receiving permission from the previous journal is completely inappropriate. Nowhere in this manuscript do they state that this data has already been published; rather it is presented as new unpublished data. This represents a serious ethical violation of data duplication between manuscripts and/or self-plagerism.
Minor points
1. The authors did not add labels to the figure panels to clearly indicate which images are from CT, HS and CAF mice, as I previously suggested. They did this in their 2018 paper, so there is no reason they can’t do the same thing in these figures.
2. The authors need to be more careful when they discuss significant differences in weight between the CT and HS groups in Figure 1. In their Reviewer Response, the authors state ‘…there are no significant differences in weight between CT and HS groups’. While this is true at the 12 week time point, the authors own analysis indicates that there is a significant difference in weight between CT and HS at the 4 week timepoint. The authors need to clearly define this distinction, specifically that there is no significant effect at the endpoint (12 weeks). This needs to be clearly defined in lines 412-414 and all other similar sections.
3. Astrocyte analysis in Figure 13: The authors still do an insufficient job of demonstrating how they analyzed astrocyte branches. They should show example(s) of higher magnification, single astrocytes, showing their different branches/arbors. I think showing the processed/analyzed images as is done in reference 104 would be extremely helpful for readers.
Author Response
Please see the attachment.

Reviewer 2 Report (New Reviewer)
Comments and Suggestions for Authors
In the manuscript by Mota et al., authors investigated the effect of two types of carolic diet on the hippocampal GABA-ergic system in young rats. Authors observed changes in the subpopulation of GABAergic neurons, altered expression of reelin, and region-dependent changes in morphological parameters of astrocytes. Authors link these alterations to the previously observed behavior changes in rats supplemented by the same diets. The manuscript is well written with a detailed Method section, and an informative introduction. Results are presented in a structured way and clearly written. Conclusions are supported by the obtained results.
Minor comments:
The figures located between lines 153-154, 178-179, 202-203, 225-226, 249-250, 273-274 are misleading. Are they from the previous edition of the manuscript?
Line 160: The sentence starting from “Note that CAF-treated induced…” should be revised. Perhaps the author mean to say: Note that CAF-treated animals showed a significant …
Line 342-343: The sentence “These length distributions were unchanged for the higher length distributions in both regions.” is confusing. Consider modifying it to make it more clear.
Comments on the Quality of English LanguageGood level
Round 2
Reviewer 1 Report (Previous Reviewer 2)
Comments and Suggestions for Authors
The author's have removed Figures 1-2 and altered the text accordingly.
However, I find their comments in the Response to Reviewer about this issue to be completely underwhelming and they indicate that the authors do not appreciate the seriousness of their offense: Replicating exact figures from a previous publication without proper acknowledgement AND without permission from the journal is a serious breach of ethical norms and publication standards. I would hope the authors would not make such a blatant disregard of these important issues in the future.
Author Response
Dear Reviewer,
Thank you for your comment. As mentioned in the previous round of revisions, we did not deliberately have any intention of self-plagiarism of the weight results. However, we agree with the Reviewer that we should have paid more attention to this issue and, for sure, we clearly will not repeat this in the future. Thank for your comment once again.
This manuscript is a resubmission of an earlier submission. The following is a list of the peer review reports and author responses from that submission.
Round 1
Reviewer 1 Report
Comments and Suggestions for Authors
In the manuscript by Mota et al., the authors evaluated the effect of two different chronic high-caloric diets (during adolescence to adult phase) on the expression of inhibitory neuron related peptides using rats. The authors showed that high calorie cafeteria diet reduced the expression (immunoreactivity area) of PV+ and NPY+ neurons along with reduced mRNA expression of reelin, whereas high-sugar diet induced reduction of expression of PV+ neurons and an increase in CB+ neurons in specific sub-regions of dorsal hippocampus. Also, they observed an increase in the length of astrocyte processes without any changes in number of astrocytes. I appreciate the author's effort for using a chronic high calorie diet paradigm from adolescence to adult phase, however, the authors did just one experiment and included several observations using mostly immunohistochemistry and some qpcr results without any functional or behavioral relevance for such changes, which lacks my enthusiasm for this manuscript. Also, there are several concerns as below.
There is no data included to support the claims of changes in adipose tissue in the fig. 1.
It is hard to see any differences from the IHC representative images. The authors should label the immuno-positive (PV+ or NPY+) neurons with an arrow or include an inset 'zoomed' images showing the immuno-positive neurons.
It is unclear why the authors did not test the mRNA expression levels of Parvalbumin and Npy, which would have clarified if the changes observed in IHCs are due to reduced expression of neuropeptides or changes in neurogenesis?
There is no evidence of functional or behavioral relevance to these changes. The authors could have performed behavioral tests relevant to cognition/learning or memory since they hypothesized that the high calorie diets induced cognitive deficits.
It is unclear why the authors evaluated the changes in astrocytes after high calorie diets? How do the authors connect observed changes in astrocyte (lack of change in number of astrocytes but increased astrocyte branch length) to reduced IHC immunoreactivity to PV+ and NPY+ neurons?
Reviewer 2 Report
Comments and Suggestions for Authors
In this manuscript, the authors give 3 sets of rats a control diet (CT), a high-sugar diet (HS) and a high-fat, increased sugar ‘cafeteria’ diet (CAF) for 12 weeks. The authors then perform immunohistochemistry to determine how dietary changes affect interneuron subtypes, astrocyte morphology and neurogenesis in the hippocampus. While the authors do find some changes in interneuron cell numbers and astrocyte morphology, I do have some significant concerns about this study, notably why the weight changes in CT vs. HS rats do not follow the expected pattern. Additionally, the authors do a poor job of showing figures that support their findings. And much of their discussion consists of trying to link random observations of other studies with their findings, many of which display a poor understanding of biology. Overall, I think this study represents a very superficial investigation with some flawed logic in how their findings about dietary intake may relate to changes in animal behavior. My specific comments are below.
I have some general comments that apply to Figures 2-6:
1. The images are too low magnification to be useful. I understand the concept of showing the entire hippocampus, but his low resolution does not allow the reader to actually see the cells immunostained in the images. The authors need to show more zoomed-in, higher magnification images so readers can see the cells. This is even more important for cells/regions where there is a significant difference, such as PV and NPY. For SST cells, the authors should zoom in on the OLM cells, as this is the only layer/location they reside in the hippocampus.
2. Each panel should be clearly labeled CT, HS and CAF to make it easier for the reader to know which image is which. And the CA1/CA3/ML/GL labels are very small, they need to be significantly larger.
3. The graphs are too small to easily read. The font in the x-axis is way too small, most people will have trouble reading the labels. All of these graphs need to be blown up in size.
2.1 Body weight/caloric consumption section: If I understand the authors correctly, the CT rats weighed on average 25 grams MORE than the HS rats after 12 weeks (435 CT vs. 410 HS), yet the CT rats intake was ~800 kcal LESS than the HS rats (14,453 CT vs. 15,223 HS). I don’t understand this discrepancy, and the authors don’t attempt to explain it anywhere. Why are the CT rats gaining significantly more weight than the HS diet rats yet eating significantly fewer calories? Additionally, the HS rats had more adipose tissue compared to CT (lines 115-117), yet still weighed less? Something is very confusing about this data and doesn’t make sense for why the CT rats weigh more than HS mice. If this is a labeling/writing error by the authors, it is an important one that needs to be correct. If this does in fact represent the real data, then the authors have to explain this discrepancy. And more importantly, if the rats are not gaining the expected weight based on their increased sugar HS diets, then it calls into question their entire findings and logic of the experiment. So I view this finding, if accurate, as extremely problematic based on my understanding of how these different diets should affect rat weight over the 12 weeks.
Lines 69-72: The logic for using BDNF, RELN and CDK5 as ‘neurogenesis’ markers seems far-fetched. There are many other, more traditionally defined genes that are better to study neurogenesis in the hippocampus. For example, Reln is expressed by both MGE- and CGE-derived interneurons in the hippocampus, so there is no way that the authors can relate hippocampal Reln expression to neurogenesis, as opposed to changes of Reln in interneuron expression.
Lines 88-90: Need to define HS and CAF here, you don’t define these abbreviations properly in the text.
Lines 102-104 and all related statements in the Results: It is very odd that the authors do not add the measurements in the text. The authors need to state ‘435 g… 410 g… 460 g.’ Same thing in lines 108-109 and every other section below. These numbers are meaningless without having the measurement label after each number, it is standard in scientific journals to have the measurement directly after the number.
It is standard nomenclature to refer to somatostatin cells as SST or SOM. ‘SS’ is not a typical abbreviation for somatostatin.
There are excellent reelin antibodies available, can the authors explain why the performed qPCR on Reln rather than immunostaining?
Astrocyte analysis in 2.4: The authors have to show high magnification images of these astrocytes. These images in Figure 12 are useless for highlighting these differences. The authors explain how they measured astrocyte length/branching in the methods, so they should show better representations here. If these affects are as striking as they claim in the graphs, it would be great to show 1-2 high magnification images of astrocytes (with reconstructions) from CT and CAF brains.
Lines 351-353: This statement, that there were no significant differences in weight gain, is not consistent with Figure 1 and lines 104-107, where there were significant differences between some of the diet conditions. I don’t understand this apparent contradictory claim, the authors need to better explain or modify this statement.
Lines 368-370: It is a drastic overstatement to claim that no changes in GAD1 mRNA means that there are not changes in GABA release. The authors did not directly examine GABA release, so they should remove this claim as their data does not support it. The last sentence of this paragraph is more accurate, but again is based on the faulty assumption of this first statement.
Lines 441-442: The authors state that reduction in NPY could related to cell death and decreased activity, but they don’t provide any evidence for this statement. No citation. What is the basis for this claim?
Lines 458-459: It’s an overstatement to claim that the cholinergic system is not affected just because there are no changes in VAChT levels. The authors should modify this statement.
Lines 470-485: This whold section about BDNF and neurogenesis is awkward. The authors somehow integrate these BDNF changes (or lack thereof) with memory deficits, despite not looking at memory in this study. They have no basis for relating alterations in spatial learning from other studies are not related to THEIR lack of changes in BDNF (lines 483-485): this requires logical leaps of faith that are not possible. Especially in light of how often the authors accurately state that differences between studies can be due to changes in types of diets and/or length of diet administration. They have no basis for relating genetic changes in gene expression in their study to changes in behavior in other studies using different diet protocols. Maybe the rats in this study do have alterations in memory function? The authors never tested this.
Lines 500-502: The authors’ logic relating Reln to PV protein expression is flawed. Reln and PV are expressed in non-overlapping populations in the hippocampus, so how would decreased Reln mRNA affect PV levels? This is one of numerous examples where the authors take observations out of context and try to generate possible relationships and/or causal relationships between findings.
Lines 507-511: Again, the logic that decreased DCX (a marker of postmitotic neurons) could cause reduction in Reln, does not make sense. And then extending this flawed logic to contributing to spatial memory deficits is an incredible stretch of the imagination.
